# Gamified Participatory Sensing in Tourism: An Experimental Study of the Effects on Tourist Behavior and Satisfaction

**Shogo Kawanaka [1,2,3,\*]** , **Yuki Matsuda [1,3]** , **Hirohiko Suwa [1,3]** , **Manato Fujimoto [1,3]** , **Yutaka Arakawa [4,5]** **and Keiichi Yasumoto [1,3]**

[1]   Nara Institute of Science and Technology, Nara 630–0192, Japan; yukimat@is.naist.jp (Y.M.); h-suwa@is.naist.jp (H.S.); manato@is.naist.jp (M.F.); yasumoto@is.naist.jp (K.Y.)

[2]   Japan Society for the Promotion of Science, Research Fellowship for Young Scientists, Tokyo 102–0083, Japan

[3]   RIKEN Center for Advanced Intelligence Project AIP, Tokyo 103–0027, Japan

[4]   Graduate School and Faculty of Information Science and Electrical Engineering, Kyusyu University, Fukuoka 812–0053, Japan; arakawa@ait.kyushu-u.ac.jp

[5]   Japan Science and Technology Agency, Presto, Tokyo 102–0076, Japan

\*   Correspondence: kawanaka.shogo.kp1@is.naist.jp; Tel.: +81-743-72-5392

**Abstract:**   In the tourism sector, user-generated information and communication among tourists are perceived to be more effective and reliable contents. In addition, the collection of dynamic tourism information with high spatio-temporal resolution is required to provide comfortable tourism in response to the changing tourism style with the advancement of information technology. Participatory sensing, which can collect various types of information is a useful method by which to collect these contents. However, continuous participation of users is essential in participatory sensing, and it is one of the most important points to stimulate participation motivation. In the tourism situation, we also need to pay attention to the total tourist satisfaction of participants. In this paper, we adopt gamification, i.e., the implementation of game design elements in real-world contexts for non-gaming purposes, for participatory sensing as an incentive mechanism to motivate participants with active participation and collect the necessary information efficiently. Within the framework, where points are given when completing the requested sensing task (=mission), two sensing missions with different burdens; Area Mission and Check-in Mission, and three different types of rewarding mechanisms; Fixed, Variable and Dynamic Variable, are designed as a gamification mechanism. We implemented these elements in the proposed participatory sensing platform application and conducted an experimental case study with 33 participants at an actual tourist spot: Kyoto, Japan. Then, we investigate the effects on tourist behavior and satisfaction by analyzing collected sensor data, mission logs, and post-survey answers. As a result, we can conclude the following: (1) the tourist behavior is changed due to the proposed gamification design and the necessary information was collected efficiently; (2) the participants tend to prioritize Check-in Mission over the sightseeing, which can induce a behavior change but might impact sightseeing enjoyment.

**Keywords:** participatory sensing; gamification; tourism; incentive mechanism; behavior change

## 1. Introduction

User-generated tourism content is perceived as an effective and reliable information resources for other tourists. In the tourism sector, firms use crowdsourcing to exploit travelers as marketers and actively involve them in their marketing strategies [1]. In addition, social problems such as overtourism, which is defined as "the impact of tourism on a destination, or parts thereof, that excessively influences

perceived quality of life of citizens and/or quality of visitors experiences in a negative way" by The World Tourism Organization of the United Nations (UNWTO) [2], caused by an increase in the number of tourists, have recently attracted attention. Collecting crowdsourced user-generated content as well as tourist activity flows can help solve these problems. Furthermore, along with the development of information technology in recent years, a tourism style that decides the next tourist destination during sightseeing such as on-site tour has also appeared. In order to realize more comfortable on-site tour, it is necessary to collect dynamic tourism contexts, such as smoothness of pedestrian flows, crowds in mobility, temporary events and temporary closures of tourist facilities, efficiently and with high spatio-temporal resolution [3].

One method to collect such information, analysis of data from social networking services linked with geolocation information can be considered. In the research field, many analyzes using data from Twitter have been conducted, but geo-tagged tweets are reported to be less than 1% of all tweets [4–6]. Additionally, since most people don't tag their precise location in Twitter, it was officially announced that this ability is removing (https://twitter.com/TwitterSupport/status/1141039841993355264). Then, as a suitable method for accumulating such contents, participatory sensing, which collects various data associated with location information from a smart device owned by people [7], exists. The widespread use of smartphones [8] with a variety of embedded sensors, such as GPS, camera and accelerometer creates the potential for dense, high-quality participatory sensing. In addition, it is possible to collect not only information obtained from sensor data but also content that can be generated only by humans such as information using human perception and impression through experience. However, the participant must bear the burden with respect to battery consumption, memory/storage capacity, and mobile data traffic and possibly even their behavior and time. It is difficult to collect data continuously relying on only the voluntary participation of people because of these loads. In addition, it is necessary to consider the burden on time especially in tourism situations. Tourism is an invaluable time for people and sometimes it is a once-in-a-lifetime experience. They may need to spend their time to generate content during the trip, and occasionally take detours to upload content before they make their way to their next destination. The design of incentive mechanisms for participation is essential to realize continuous contributions by motivating participants.

Two types of incentives are generally discussed in participatory sensing: monetary incentives and non-monetary incentives [9]. Monetary incentives provide money as a reward for contributing to sensing. A variety of studies have been conducted to determine the impact of the price rate and reward mechanism (e.g., fixed micro-payment, variable micro-payment, lottery and auction), on the participation rate and on the quality and quantity of the collected data [10,11]. On the other hand, non-monetary incentive is a mechanism that provides intangible value, such as fun, useful information, psychological need satisfaction, and memorable experiences [12]. Among non-monetary incentives, gamification that incorporates game thinking and mechanism into non-game content has attracted attention [13–15]. Gamification has been introduced to the following areas: education [16], health care [17], marketing [18] and social networking [19]. There have been many studies on participatory sensing applying gamification, and the effectiveness of gamification has been demonstrated [12,20–24]. However, the literature suggests that empirical studies are required in order to clarify the effects of gamification in each participatory sensing context or domain [23,25–27]. Even in tourism sector, gamification has been applied to improve tourism satisfaction or to generate brand awareness [28]. However, few concrete studies have addressed the appropriateness of the design of gamified participatory sensing for tourists. That is, the purpose of introducing gamification in this study is not to improve tourism satisfaction or to generate brand awareness, but to collect dynamically changing tourism information efficiently through participatory sensing for tourists while taking into account the burden on tourists. In order to realize a sustainable participatory sensing system in the tourism domain, a detailed gamification design should be discussed.

In this paper, we adapt gamification to participatory sensing in order to efficiently collect dynamic tourism information while taking into account the burden on tourists. And we investigate the

effect of gamification on tourist behavior and tourist satisfaction through an experimental case study. We designed several gamification mechanisms, which have different types of sensing tasks and reward mechanisms, and implemented these mechanisms in ParmoSense [29], an integrated participatory sensing platform that we developed. Gamification mechanism design consists of two types of sensing tasks (referred to hereinafter as missions): the Area Mission involves for walking around a specific area, and the Check-in Mission involves taking a picture at a checkpoint. There are three types of reward mechanisms, Fixed, Variable and Dynamic Variable reward mechanisms for each mission. In addition, three types of user types (sightseeing, reward, game) were designed according to the motivation of the participant, since it was indicated that the effect of gamification differs according to the participant type [30]. In order to confirm the effects of the designed gamification mechanisms, we conducted a real-world experiment in Kyoto, Japan with 33 participants. Participants used a smartphone application for sensing during sightseeing. After the experiment, we investigate the effect on tourist satisfaction and behavior by analyzing post-survey and collected sensor data. Specifically, the main contributions of the present paper are summarized as follows:

- First, to the best of our knowledge, this is the first study to elucidate suitable gamification in participatory sensing for tourists in order to efficiently collect dynamic tourism information.
- Second, we present the design of two types of mission with different burdens (Area Mission and Check-in Mission) and three types of reward mechanisms (Fixed, Variable, Dynamic Variable). Then, we implement these elements in our participatory sensing platform application and conduct a subjective sightseeing experiment in the real world.
- Third, we confirm that the tourist behavior is changed due to the proposed gamification design and that the necessary information was collected efficiently from the quantitative evaluation by analysis of collected sensor data and the statistical results of the post-survey.
- Finally, summarizing the results of the present study suggests that suitable gamification to collect tourist information efficiently, considering tourist satisfaction, is achieved by the Area Mission with the Variable reward mechanism and Free posting.

The rest of this paper is organized as follows. Section 2 reviews existing work related to this paper and sums up the challenges of the present study. In Section 3, we describe the proposed gamification mechanism design and implementation of the application. Then, we mention our settings of the subjective sightseeing experiment in Section 4. We present the experimental results in Section 5, and a discussion and the limitations of the present study are provided in Section 6. Finally, Section 7 concludes this paper.

## 2. Related Work and Challenges

### 2.1. Incentives in Participatory Sensing

Participatory sensing is leveraging for various domains such as environmental monitoring [31,32], road condition monitoring [33,34], and health monitoring [35,36], through the widespread use of smart mobile devices which are equipped with various built-in sensors such as GPS, camera, accelerometer. Furthermore, various platforms [29,37] and frameworks [38–40] have been developed with the goal of facilitating participatory sensing in various situations. On the other hand, since the processes of participatory sensing require the participants to use a smartphone, problems, such as consumption of batteries and mobile data traffic, security of data, and quality of data, are encountered by the participants [41]. Active participation of participants is one of the most important factors in collecting data in participatory sensing. Therefore, many studies on the incentive mechanism have been conducted in order to encourage active participation [21,42]. Incentive mechanisms are generally classified into two types: monetary incentives and non-monetary incentives [43].

Many studies have investigated the effect on sensing participation and the quality and quantity of data obtained from participants introducing monetary incentives in crowdsourcing participatory sensing.

Lee et al. [10] proposed a dynamic pricing system for participants based on reverse auctions in which sensing participants set a price when providing data to the organizer (provider). Simulation analysis shows that the proposed method reduces incentive costs and improves incentive fairness. Khoi et al. [11] proposed three types of financial incentive schemes (e.g., lottery-style, fixed micro-payment, and variable micro-payment), and case study experiments in the real world with 230 participants showed that monetary incentives improve the participation rate. However, monetary incentives in the real world are often limited by total budget and have continuity problems.

On the other hand, non-monetary incentives provide experiences, such as fun [44], social contribution, and intrinsic motivation [45]. Gamification, which is defined as "the use of game design elements in non-game contexts" [13] has attracted attention in user-centric systems such as participatory sensing. Gamification refers to design that seeks to, first, increase the motivation of users or participants to engage in an activity or behavior and, second, to increase or otherwise change a given behavior [23]. For instance, Palacin-Silva et al. [27] introduced gamification (e.g., storytelling, challenges, points, and leaderboards) to the environmental sensing domain in order to explore the impact of gamification on engagement and user experience. Through an experiment, they found that gamification affected participant engagement in a positive way (producing more submissions) without improving or compromising the user experience. Berkel et al. [22] adapted points, leaderboards, time restriction and visual feedback as gamification elements to measure the difference in the quality and quantity of responses. Ueyama et al. [20] introduced a novel incentive mechanism employing gamification elements such as points, rankings, and badges, in addition to monetary rewards, to suppress a rapid rise in monetary rewards. The results show that monetary incentives are suppressed and the participation rate is increased by gamification. According to surveys and reviews on designing effective incentive mechanisms, points, rankings/leaderboards, and achievement are often introduced as gamification elements, and affect the user positively, particularly in participatory sensing [23,43,46].

As mentioned above, previous research has shown that both monetary and non-monetary incentives increase the participation rate of participants and improve the quality and quantity of the data. However, it has become clear that, especially with respect to monetary incentives, the imbalance between tasks and rewards can cause a decline in the quality of data, and the influence on the control of user withdrawal decreases in long-term experiments (e.g., monetary incentives are exhausted during the campaign) [23,47]. Therefore, monetary incentives should be implemented cautiously in combination with gamification [23,48].

## 2.2. Gamification in Tourism

Gamification is often introduced in the tourism domain in order to raise brand awareness, enhance tourist experiences, destination loyalty, consumer loyalty and engagements [1,18,26,28,49–53]. One of the successful examples of gamification in tourism is TripAdvisor (https://www.tripadvisor.com/), the world's largest travel portal service. It has several website tasks which use various gamification elements. For example, tourists are motivated to upload tourist information such as photos and/or reviews and are presented with points, badges, competitive scoreboard and so on [54]. Moro et al. [55] have shown that some badges and the total amount of badges acquired are the most relevant gamification feature for motivating tourists to write reviews.

In the context of brand awareness, Foursquare (https://foursquare.com/) has successful partnerships with many brands which are promoted during game play through check-ins and when sharing their experiences through social media. In addition, there are games based on Facebook, such as Ireland Town and Smile Land Thailand game (http://smilelandgame.com/), which generate brand awareness as a tourist destination and increase user frequency of social networking media of local associations. For tackling the problem of waste generated in tourist attractions, a gamification application called Wastapp, has been developed to promote recycling behavior of

tourists. In experiments using this gamified application, it has been reported that gamification contributes to the recycling behavior of tourists and improves the image of destinations [56].

Fully-fledged games, which are more about fun and entertainment, also exist to encourage tourism. Geocaching (https://www.geocaching.com) is an outdoor recreational activity, in which participants use GPS-enabled devices, such as smartphones, to find treasure boxes on site. Pokémon Go (https://www.pokemongo.com/) is one of the most popular augmented reality (AR) games thus far. In the Pokémon GO game, players must catch and fight Pokémon while exploring and experiencing the real world [57]. These were not specifically designed for tourism. However, Geocaching partners with local tourism associations to create special treasure hunt tours and partnering Pokémon GO partners with UNWTO to develop innovative tourism experiences through real-world games (https://www.unwto.org/global/press-release/2018-11-20/unwto-partners-niantic-develop-innovative-tourism-experiences-through-real-). In both games, players look for hidden treasures or Pokémon at various real-world locations. Through this engagement, players are encouraged to interact with the destination at site [51].

Gamification is widespread in the tourism industry as stated previously. However, according to Xu et al. [51], academic research on the application of gamification specifically in the tourism field is still scarce. There are a few researches which investigating the impact of gamification on dynamic tourism information collection with various gamification mechanics. However, although a comprehensive evaluation using various mechanics was conducted, it is not clear which gamification mechanics had an impact on which outcomes [53].

## 2.3. Challenges

There have been many studies on motivation to encourage the active participation of users in participatory sensing. In particular, the utility of gamification in user-centric services such as participatory sensing has become clear. However, the literature suggests that empirical studies are required in order to clarify the effects of gamification in each participatory sensing context or domain [23,25–27]. Gamification is often used to raise brand awareness, and enhance tourist satisfaction and engagement in the tourism domain. Even TripAdvisor, which applies gamification actively in the tourism sector, focuses mainly on exchanging and posting tourist information before or after sightseeing [54,55]. Few academic studies aim at efficient collection of dynamic tourism information with high spatio-temporal resolution by participatory sensing during sightseeing. In the case of crowdsourcing sensing tasks to tourists during sightseeing, participants are burdened by spending not only their device resources but also invaluable time for sightseeing. It is necessary to create a framework which tourists can easily follow and do even during sightseeing, in order to realize efficient dynamic tourism information collection.

Our research question is what is the suitable gamification in the participatory sensing for tourists considering tourist satisfaction. An experimental case study is conducted in order to clarify this research question.

## 2.4. Preliminary Work

In the preliminary study, we described the overall design of the study and investigated the effect of gamification design on tourist behavior and satisfaction using only the statistical results of a post-survey [58]. In this paper, we mainly report more behavioral analysis results. We conducted quantitative and qualitative evaluations using the logs and sensor data obtained from smartphones through experiments. The results and additional discussion further supported the validity of the results obtained in previous studies.

## 3. Study Design

In this paper, we clarify the gamification design that is suitable for user participatory sensing with tourists as participants in order to efficiently collect tourist information without degrading tourism

satisfaction. Consumer-generated media (CGM) required for tourism is inertial sensors and GPS built into smartphones, and photos and comments. With the development of human activity recognition research, it is possible to collect information such as tourist behaviors [59] and congestion degree in the surrounding area [60] from sensor data collected from smartphones. Also, photos and comments are very useful information for the next tourist to understand the current situation of the sightseeing spot.

Some of the most commonly used gamification mechanics in participatory sensing or location-based social networks (LBSNs) are points, rewards, levels, badges, leaderboard/ranking, missions and so on [23,43,46,61]. Jordan [62] analyzed how gamification mechanics, especially mayorships and budges, in Foursquare which is one of the most famous LBSNs, can impact people's mobility decisions. Mayorship was reported to be highly effective in encouraging multiple visits to the same location. It has also been stated that Badge has the effect of influencing a person's mobility to gain a new badge and explore physical space in some active users. Molo et al. [55] have shown that several specific badges and the total amount of badges acquired are the most relevant gamification feature for motivating tourists to write reviews useing TripAdviser's reviewing data. Lee [53] investigated the impact on tourism behavior in cultural heritage cruise tourism using app with gamilication mechanics such as quizzes, avator, and missions. As a result, the effectiveness of gamification apps in improving understanding of cultural heritage was verified. However, although a comprehensive evaluation using various gamification mechanics was conducted, it is not clear which gamification mechanics had an impact on which factors. Since gamification designed for the collection of dynamic tourism information during sightseeing has not been explored in detail so far, we focus primarily on simple gamification mechanics, missions and point-based rewards.

The basic design assumes that each time participants perform a mission requested through the app, the participants are given points in the application. The detail of mission should be designed based on our requirements of tourism information to be collected (i.e., photos, comments, and sensor data) and the acceptability of mission. We designed two kinds of missions with different burdens on participants in order to clarify a suitable gamification design. In order to efficiently collect the information that campaign organizers (e.g., municipality or tourism association) require, we hypothesized that this could be achieved by changing the reward mechanism according to the demand level of information and designed three reward mechanisms. The following explains the details of each mission design, reward mechanism, and an outline of the application that implements these elements.

### 3.1. Mission Design

The priority of sightseeing will vary depending on the burden of the mission on the participants. Here, the priority of sightseeing means a degree of concentration against sightseeing during the mission that the participant performed. The purpose of tourists is obviously sightseeing. Therefore, the mission should not interfere with sightseeing and the burden of the mission must be low.

We designed two types of missions with different workloads for the participants, namely, an Area Mission, which can be implicitly feasible and low-load, and a Check-in Mission, which requires performance explicitly and is high-load. In addition, we designed Free posting, which the participants can perform anytime and anywhere.

**Area Mission**

The Area Mission is displayed as multiple polygons in a specific sightseeing area on the map, as shown in Figure 1. Sensor data (GPS, acceleration, angular acceleration, geomagnetism, and illuminance) are collected by strolling in the displayed area. These sensor data can be used to estimate the context of sightseeing attraction such as the smoothness of pedestrian flows [60], the congestion on the roadway while traveling with a vehicle [63], and the detailed sightseeing behavior of the user [59]. Points are given at fixed time intervals, by collecting sensor data. A gold area, a silver area, and a bronze area are set according to the points to be given.

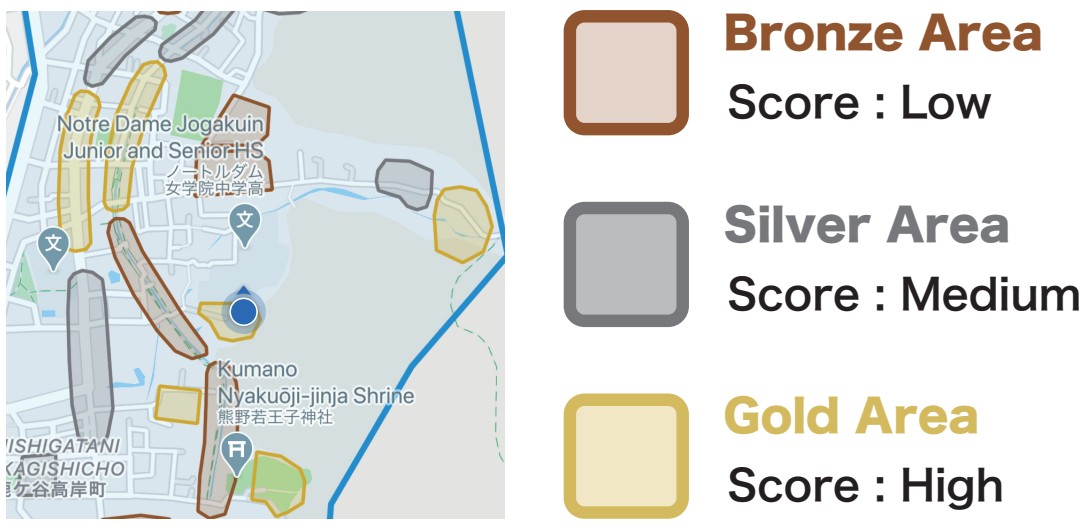

**Figure 1.** Area Mission.

**Check-in Mission**

The Check-in Mission is displayed with a pin at a specific sightseeing spot on the map, as shown in Figure 2. It is possible to check in when the user is within a certain distance from the pinned place. By posting photographs and comments on the spot, check-in is completed and points are given. Temples/shrines, museums, restaurants, and souvenir shops, which are commonly mentioned as sightseeing spots, are set as types of pins. Photographs and comments can also be used to estimate the context of tourist site by analyzing these data. In addition, it is possible to directly convey the state of the sightseeing spots. The colors of the pins are set according to the points, similar to the Area Mission.

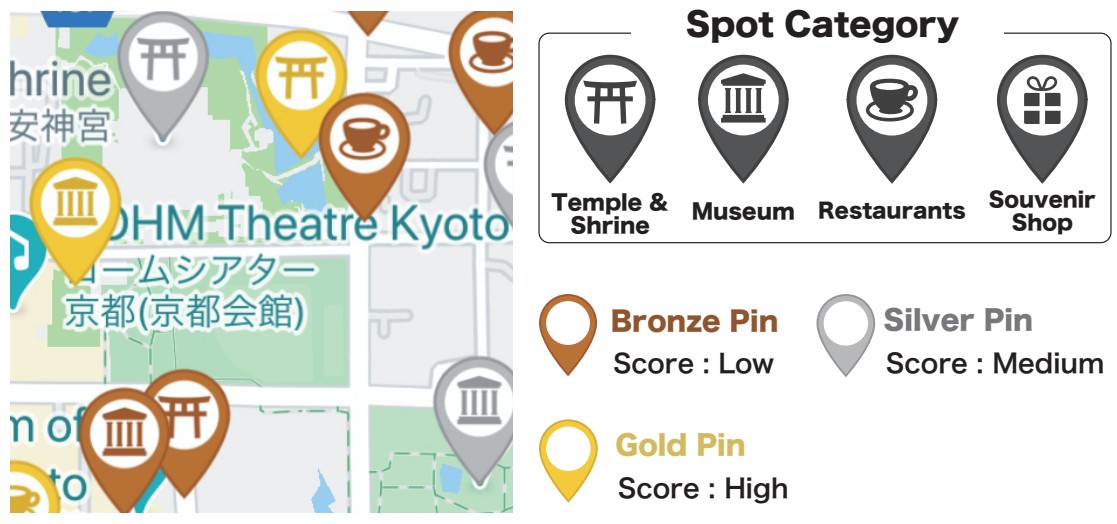

**Figure 2.** Check-in Mission.

In addition, *Free posting* is also designed so that the users can freely post photographs of places they find interesting and share them with other tourists. These photographs and comments posted for the Check-in Mission and Free posting will be shared with other tourists through the timeline function shown in Figure 3(6). The characteristics of each mission are summarized in Table 1.

**Table 1.** Summary of mission characteristics

|  | **Area Mission** | **Check-In Mission** |
|---|---|---|
| Sensing style | Implicit | Explicit |
| Data collection ability | Low | High |
| User burden | Light | Heavy |
| Sightseeing interference | Low | High |

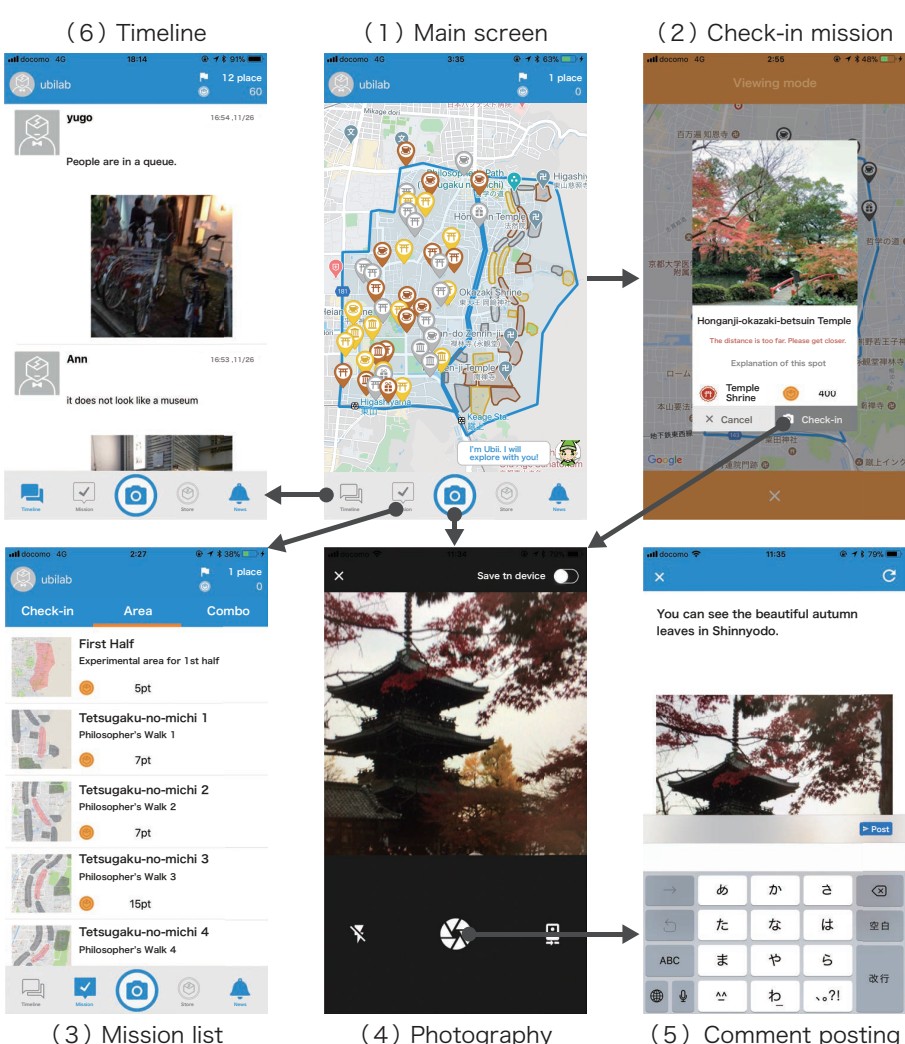

**Figure 3.** Parmosense's application screens and transition pages.

*3.2. Reward Mechanism*

In participatory sensing, the necessary amount of information and the responsiveness required by the organizer varies according to the spot. Using the concept of dynamic pricing, we hypothesize that by changing the points assigned according to the demand for information, it is possible to change the behavior of tourists and collect necessary information [64]. We designed three types of reward mechanisms as follows.

**Fixed reward**

Fixed rewards are obtained depending on the type of mission (Area Mission, Check-in Mission).

**Variable reward**

The rewards vary by spot in the case of Variable reward mechanism. We assume that the event organizer sets the information demand for each spot, and the reward is decided according to the balance between the demand and the supply. In the experiments described in Section 4, the number of hits when each sightseeing spot name was searched on the web was assumed to be the demand of information, and the points were determined accordingly. For instance, famous sightseeing attractions already have a lot of information on the web and fresh data may also be supplied naturally because many people visit such sites. Therefore, the demand level of information will decrease and fewer points are to be gained for such places. On the other hand, not well known sightseeing attractions are assigned high point values. This is because the amount of information on the Web and the amount of information supplied are assumed to be comparatively small.

**Dynamic-variable reward**

In addition to the Variable reward, the amount of information collected by participatory sensing is reflected as the demand level of information at fixed time intervals. For example, suppose that a high score is set for an unknown spot, and many tourists posted at that spot in a short period of time. The next participant who completes the mission at that spot will receive a lower reward. On the other hand, if the data is not updated for a long time even for a famous sightseeing attraction, a high score will be assigned. For the sake of simplicity, we set the weight in advance and change this weight every 30 min accordingly in the experiment.

*3.3. ParmoSense*

Our smartphone application for participatory sensing is called *ParmoSense* [29] and consists of six screens, as shown in Figure 3(1)–(6). The details of these screens are described as follows.

(1) This is the main screen of the application that indicates sensing tasks with pins or polygons as missions. The ranking and points of a user are displayed in the upper-right corner of the screen.
(2) This screen is displayed when the user taps a mission pin on the map. The screen shows details of the Check-in Mission. Check-in is allowed only when the user is within a certain distance from the pinned place.
(3) This screen is displayed when the user taps the mission button at the bottom of the screen. In this screen, the details of missions in the map are shown in list form.
(4) This screen is displayed when the user taps the check-in button in (2) to capture and upload a photograph or taps the camera button at the bottom of the screen for Free posting.
(5) This screen is displayed after taking a photograph in (4). The user can input texts on information or impressions of the photograph (spot).
(6) This screen is displayed when the user taps the timeline button at the bottom left corner of the screen. This screen shows photographs and comments posted by other users.

The timestamp, GPS, acceleration, gyroscopic, geomagnetism, and illuminance values of the smartphone are collected at a sampling rate of 10 Hz, while this application is running (even in the background). These data are transmitted to the server every 5 s. Sensor data are also collected at the moment when the user takes a photo and is sent to the server along with the captured photograph, independent of the periodic sensor data.

**4. Sightseeing Experiment**

In this section, we describe the sightseeing experiment. We conducted the sightseeing experiment using the developed application in order to investigate the effect of our designed gamification. In the

following sub-sections, we explain in detail the participants, the experimental procedure, and the analysis method.

## 4.1. Participants

Participants were recruited from our university by e-mail which described the purpose, details, and remuneration. A total of 33 participants (25 male and 8 female) were recruited. Most of the participants were in their 20 s (one participant was in his 30 s and one participant was in his 40 s). The number of Japanese and non-Japanese participants were 29 and four respectively.

It is important to investigate the influence of each user type by setting the user attributes in gamification [26]. Bartle's classification is a well-known classification of gamers types [30]; Achievers: people who are satisfied with achieving quests in the game, Explorers: people who explore and gain satisfaction from discovery or thrill, Socializers: people who gain satisfaction from social aspects such as interaction with others and Killer: people who gain satisfaction by competing with others. Based on these models, we set up three types of users based on the following questions; "While participating in a stamp-rally that gives a point-based reward, what do you think the most important among the following?"—Enjoy sightseeing (Sightseeing type), Enjoy stamp-rally (Game type), or Attempt to obtain more reward (Reward type).

According to the results of a pre-survey, 17 participants are classified as the Sightseeing type, seven participants are classified as the Game type, and the remaining nine participants are classified as the Reward type. The participants are assigned to these three groups taking into account the gender, nationality, and user type. The size of each group is 11 and the means and standard deviations (SD) for the number of males, Japanese, each user types in each group are follows. #Males: means = 9.67; SD = 0.58, #Japanese: means = 8.33; SD = 0.58, #Sightseeing type: means = 5.67; SD = 0.58, #Game type: means = 2.33; SD = 0.58, #Reward type: means = 3.00; SD = 0. Different reward mechanisms are applied to the groups: a Fixed reward for Group A, a Variable reward for Group B, and a Dynamic Variable reward for Group C.

## 4.2. Experimental Procedure

The experiment was conducted in Kyoto, Japan in November 2017. In this experiment, we asked the participants to perform sightseeing in an area of Kyoto City while earning points by clearing missions. After the experiment, we collected questionnaires from the participants. In order to clarify the effect due to the difference in mission type, we requested participants to engage in Area Mission and Check-in Mission separately in the first and second halves of the experiment, respectively. The experiment time was set to be 4.5 h in total, which consists of a 2.5-h course and a 2-h course planned with reference to the sightseeing model course. Ahead of the experiment, we asked participants to install our developed application on their smartphone. After that, we fully explained the usage of the application and the contents of each mission for each group. We asked participants to travel alone and on foot during the experiment. We paid each participant 5000 yen as a basic participation fee which included transportation to the venue and lunch. Furthermore, as a monetary incentive, we informed participants that they would be paid additionally from 0 to 2000 yen, depending on the ranking of the points in each group.

**First-half experiment (Area Mission)**

In the first half of the experiment, Area Missions were assigned to the participants. The course started from Keage Station to Ginkakuji Temple. The participants were asked to freely sightsee using our application. Points were given to each participant based on the following rules. These were set in consideration of the uniformity of the maximum points that can be obtained in each group in the first half and the second half.

(A)    Obtain 10 points every 10 s within the experiment area for the first half of the experiment.
(B)    Obtain 15, 10, or seven points every 10 s within special areas, such as gold areas, silver areas, or bronze areas, respectively.

(C)     Special areas are updated every 30 min.

The total number of special areas in this experiment was set to be 33 (11 areas for gold, silver, and bronze, respectively). In addition, 30 points are given for Free posting.

**Second-half experiment (Check-in Mission)**

In the latter half of the experiment, Check-in Missions were assigned to the participants. The course started from Ginkakuji Temple and ended at Higashiyama Station. The participants were asked to freely sightsee using our application. Points were given to each participant based on the following rules:

(A)     Obtain 400 points at any check-in spot.
(B)     Obtain 730∼620 points, 370∼310 points, or 180∼150 points for checking in with a gold, silver, bronze pin, respectively.
(C)     Special check-in spots with colored pins are updated every 30 min.

In this experiment, we set 45 special spots; 23 spots for temples or shrines, seven spots for museums, four spots for souvenir shops, and 11 spots for cafes. In addition, the highest number of points for all groups was set to be constant. Similar to the first half, we decided to give 30 points for each text posting.

*4.3. Experimental Hypotheses*

In order to clarify the effect of gamification factors and user type settings that we described above on tourist behavior and satisfaction,these hypotheses are summarized as follows:

**Hypothesis 1 (H1).** *The priority of sightseeing varies depending on the burden of the mission. Participation in sensing imposes a burden on resources such as devices, time, and behavior, and the priority of sightseeing will differ depending on these burdens. Therefore, we designed missions with different burdens on sensing participation, as shown in Section 3.1. Area Mission in the first half of the experiment and the Check-in Mission in the second half are conducted, and the effects on mission selection and sightseeing behavior are compared.*

**Hypothesis 2 (H2).** *The efficiency of data collection depends on the reward mechanism. The necessary amount of information and the responsiveness required by the organizer varies according to attractions. Using the concept of dynamic pricing, we hypothesize that by changing the points assigned according to the demand for information, it is possible to change the behavior of tourists and collect necessary information.*

**Hypothesis 3 (H3).** *Priority of mission and sightseeing differ according to user type. The priority between mission and sightseeing (=extent of contribution to sensing) will differ, depending on the motivation for sensing participation. User type of participant is set according to the participation motivation by the pre-survey, and which of mission and sightseeing is prioritized for each user type is compared from the post-survey.*

*4.4. Analysis Method*

Mission logs, location data from the smartphone application, and the post-survey were analyzed to verify our experimental hypotheses. First, we quantitatively analyze the effect of different reward mechanisms on tourist behavior and mission selection using mission logs (for H2). For the Area Mission, the duration of stay according to the demand is used as an evaluation index. For the Check-in Mission, the number of check-ins with respect to the demand was used as an evaluation index. In addition, the amount of information collected in each mission was evaluated from photographs and comments obtained at check-in and Free posting. Next, the effect on the route selection is qualitatively evaluated by visualizing the travel route of the participants using the location data (for H2). Finally, we investigate the priority of mission and sightseeing through a post-survey in order

to confirm the effect of participation in gamified participatory sensing (for H1 and H3). The details of the post-survey are as follows: Q1. Which did you prioritize in the first half (Area Mission) of the experiment, sightseeing or the mission?; Q2. Which did you prioritize in the second half (Check-in Mission) of the experiment, sightseeing or the mission ? These questions were answered using a five-point Likert scale in which 1 = prioritized sightseeing, and 5 = prioritized the mission.

## 5. Results

In this section, we explain the experimental results. Thirty-three participants were divided into three groups of 11 participants, each with different rewarding methods. In the first half, the participants performed the Area Mission, and in the second half the participants performed the Check-in Mission. During both halves the participants engaged in Free posting. After the experiment, we conducted a post-survey about the priority of the mission and sightseeing.

Through the experiment, approximately 830 MB of sensor data (e.g., location and accelerometer), 1744 photos and comments (688 from the Check-in Mission and 1056 from Free posting), and 33 post-survey results were collected. In order to clarify the suitable gamification for tourism, these data were used to analyze the effects of each gamification element on tourist behavior and tourist satisfaction according to the indicators shown in Section 4.4.

### 5.1. Duration of Stay and Number of Check-in

First, the duration of stay and number of Check-ins for each tourist attraction is calculated using the collected location data, and the effect on tourist behavior is clarified quantitatively. Since the experimental groups are determined by the reward mechanism, the effects of the reward mechanism on tourist behavior are clarified based on the results.

We calculated the duration of stay according to the information demand level for each group. The area according to the demand level set to Group B was used as a baseline, and this was also applied to Group A. In Group C, since the demand level changes every 30 min, the duration of stay is calculated for each demand level for each period and their sum is then calculated. Here, we cannot simply compare these groups, since the square-meter area is different for each area. Thus, we decided to use the duration of stay per square measure ratio as an evaluation index. Figure 4 shows the duration of area/square measure ratio by group.

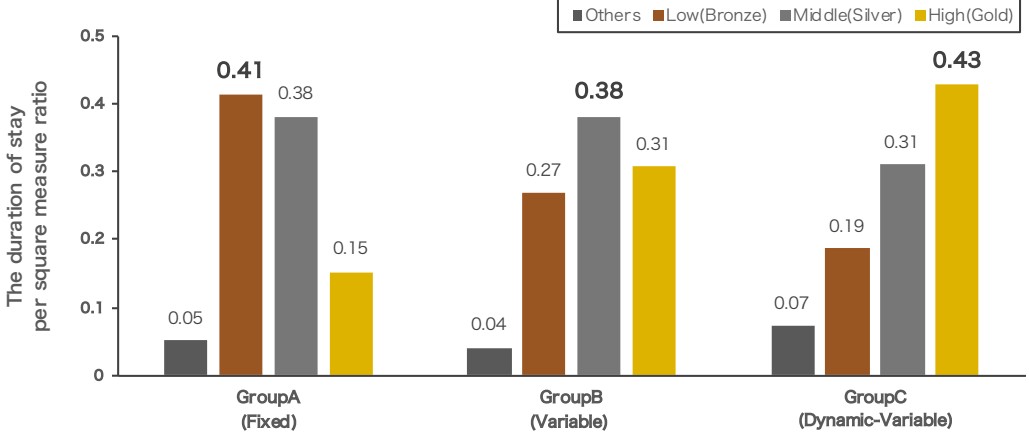

**Figure 4.** Duration of stay in each area per square measure for each group.

The duration of stay in area with low information demand level accounted for more than 40% in Group A. On the other hand, the duration of stay in the area with high demand remained approximately 15%. It is a natural outcome that in Group A, in which a fixed point is obtained at any of the spots, the duration of stay at a well-known sightseeing spot becomes the longest. In Groups B and C,

where the information demand level is indicated by the color, it is seen that the duration of stay at less popular sightseeing areas increased compared to the famous sightseeing area.

In the same way, the number of check-ins according to the information demand level is calculated for each group. The demand level of Group B was used for both Groups A and B, and that of Group C was changed for each period. Since the number of check-in spots is different between Groups A, B, and C for each demand, the number of check-in spots is normalized by the coefficient, so that the number of check-in spots is the same. The ratio of check-in spots by group is shown in Figure 5. In Group A, since there is no difference in rewards for each spot, check-in at the spot with low information demand occupied approximately 40%, and a result similar to that for the Area Mission was obtained. On the other hand, Groups B and C had the highest number of check-ins at spots with high information demand.

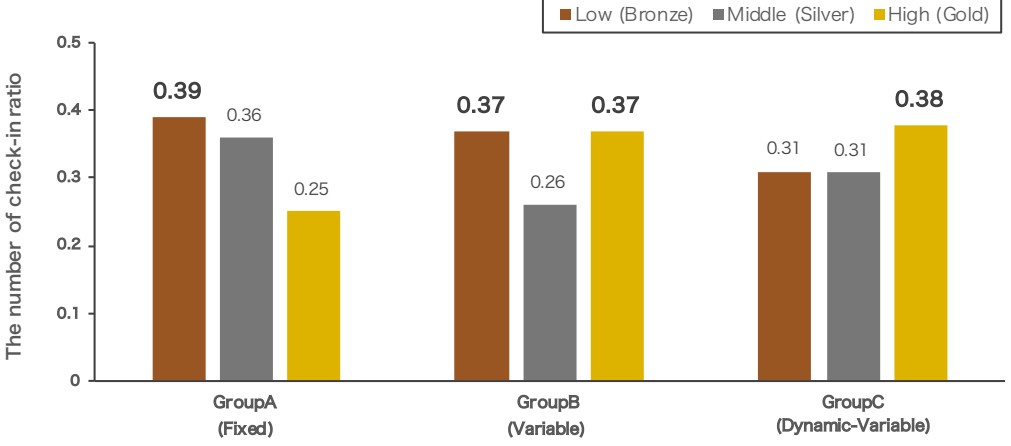

**Figure 5.** Check-in ratio for each group.

As a result, it was suggested that a Variable reward according to information demand can change the behavior of tourists and collect necessary data for both the Area Mission and the Check-in Mission. In addition, it was also found that users can accept and change their behaviors, even when there are dynamic changes in the demand level at areas or spots, as in Group C. Hence, the Hypothesis 2 (H2) is accepted through quantitative evaluation.

### 5.2. Visualization of Tourist Behavior

The difference according to the reward mechanism is qualitatively evaluated by visualizing location information collected during the experiment and comparing the information per groups. Figure 6 shows travel routes visualization of Groups A and B. Group C is difficult to represent on paper because the area and check-in spot change dynamically. Therefore, please refer to the video from the following URL (Route visualization video: https://drive.google.com/drive/folders/1sbVwEsEf3iaWhBFtxCZxF7gIG7fBc7kk?usp=sharing) (You can see the route visualization video of Groups A and B as well).

In the first half of the experiment, Area Mission, there is an obvious difference indicated ①–③ in the figure. None of the 11 people chose that route in Group A with a fixed reward. However, multiple participants of different user types selected the route indicated by a gold area where high scores could be obtained. The right-hand side of the two parallel streets indicated by ③ in Figure 6, is famous as a tourist attraction called Tetsugaku-no-michi (Philosopher's Walk). In Group A, most of participants chose Tetsugaku-no-michi on the right, while in Group B, several participants chose the ordinary road on the left-hand side.

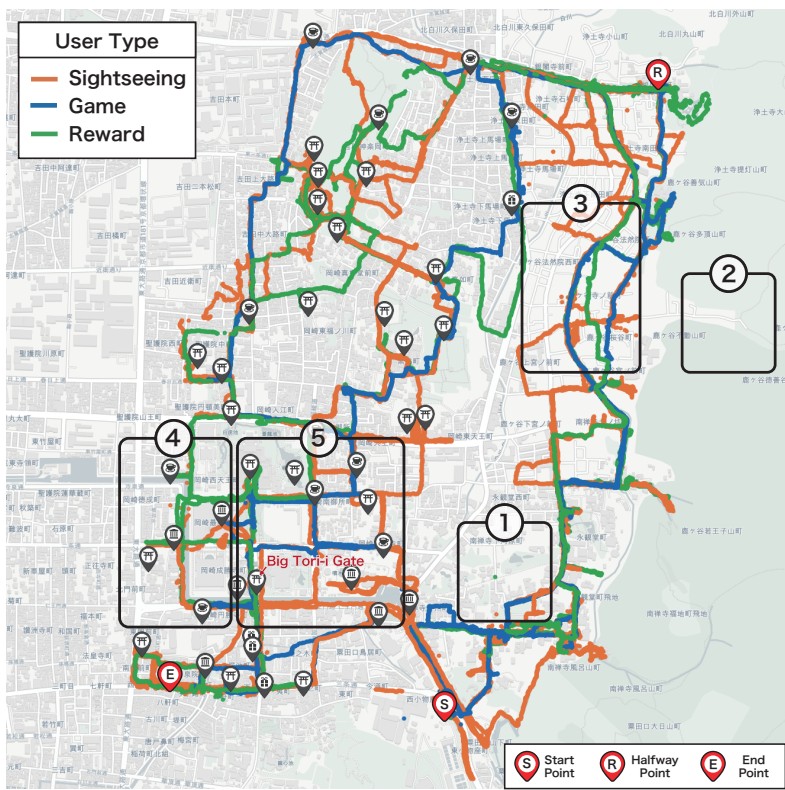

(**a**) Group A (Fixed reward).

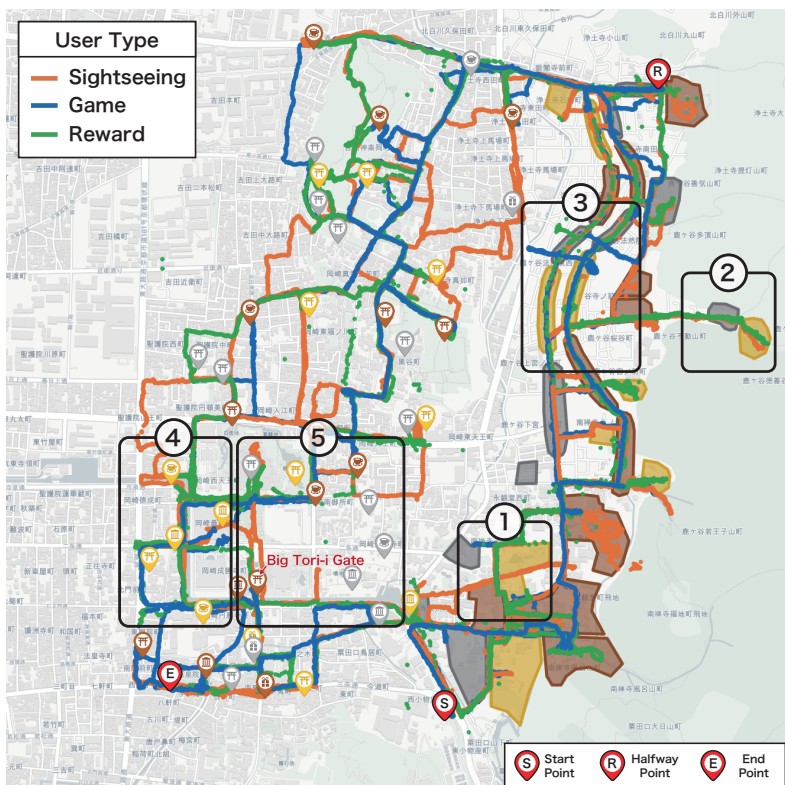

(**b**) Group B (Variable reward).

**Figure 6.** Travel routes visualization of participants. Each area indicated with ①–⑤ represent areas where obvious differences have been observed.

In the latter half of the experiment (Check-in Mission), there is an obvious difference indicated ④ and ⑤ in the figure. We can see that the participants selected the route by dispersing into the routes of ④ and ⑤ in Group A. Meanwhile, in Group B, most of the participants regardless of user type, chose route ④ in which the gold pins are located densely. The number of check-ins to the gold pins at the area indicated by ④ was compared between Groups A and B. The average numbers of check-ins were 3.8 and 7.8, respectively. We also compared the number of check-ins at Daitorii of Heian-jingu Shrine, which is a well-known tourism attraction indicated by ⑤. All 11 people of Group A checked in, while, in Group B, only five people checked in to this spot.

Even in qualitative evaluation with route visualization, we found that the variable rewarding according to the information demand level can induce tourist behavior and efficiently collect highly demanded information. Hence, the hypothesis 2 (H2) is accepted as well as quantitative evaluation.

### 5.3. Post-Survey

We investgated whether participants prioritized either sightseeing or the mission via the post-survey questionnaire. We used this information to validate the effect of gamified participatory sensing on tourist satisfaction. The results of one-way analysis of variance (one-way ANOVA) between answers to Q1 (Area Mission) and answers to Q2 (Check-in Mission) are shown in Figure 7. The average scores for sightseeing and mission priorities were 2.39 and 3.82 for the Area Mission and Check-in Mission, respectively. There was a significant difference between the Area Mission and the Check-in Mission ($p < 0.0001$). Participants prioritized sightseeing over missions in Area Mission, and prioritized missions over sightseeing in Check-in Mission. Hence, the Hypothesis 1 (H1) is accepted. Next, in order to reveal whether the priority of missions and sightseeing changes according to the differences in user type, one-way ANOVA was performed on the answers for each user type. Figure 8 shows the result, and the average scores for sightseeing type, game type and reward type were 2.82, 3.14 and 3.61, respectively. There was no significant difference between user types ($p = 0.18$). Hence, the Hypothesis 3 (H3) is rejected.

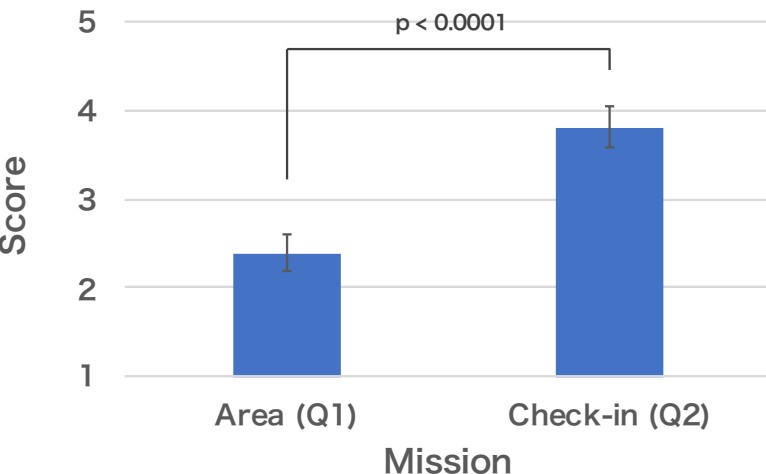

**Figure 7.** Post-survey results on priority between sightseeing and mission grouped by mission (5 = Highly prioritized mission, 1 = Highly prioritized sightseeing). Error bar represents one standard error around the respective mean value of each mission.

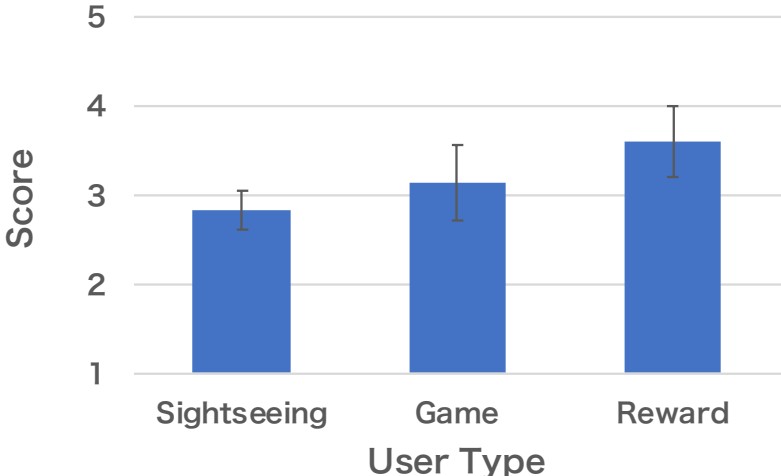

**Figure 8.** Post-survey results on priority between sightseeing and mission grouped by user types. Error bar represents one standard error around the respective mean value of each user type.

*5.4. Summary for Experimental Hypotheses*

Here, we summarize the answers for our experimental hypotheses based on the qualitative and quantitative evaluation results shown in Section 5.1–5.3.

H1: we assumed that the priority of sightseeing varies depending on the burden of the mission and it was evaluated with post-survey. The answers to post-survey on priority between sightseeing and mission were grouped by each mission type, and performed one-way ANOVA. As a result, we could find the significant differences between Area mission and Check-in mission ($p < 0.0001$) . Consequently, Hypothesis 1 (H1) is accepted.

H2: we assumed that the efficiency of data collection will depends on the reward mechanism, and it was evaluated with mission logs and location data. The impact of different reward mechanisms (experimental group) on mission choice and tourist behavior for each mission was quantitatively compared using mission logs. In addition, the impact on tourist behavior was qualitatively evaluated by visualizing the location data. In the quantitative evaluation, we found a tendency to select missions with higher scores in the Variable Reward and Dynamic-Variable Reward for both Area mission and Check-in mission. In the qualitative evaluation through visualization, we also found a change and diversification of tourist behavior in several tourist areas. Consequently, Hypothesis 2 (H2) is accepted.

H3: we assumed that the priority of sightseeing will differ according to user type i.e., motivation to participate and it was evaluated with post-survey. The answers to post-survey on priority between sightseeing and mission were grouped by each user type, and performed one-way ANOVA. As a result, we could not find the significant differences among user types ($p = 0.18$) . Consequently, Hypothesis 3 (H3) is not accepted.

## 6. Discussion and Limitations

In this section, in order to derive a clearer conclusion in the present paper, we discuss the design of gamification suitable for participatory sensing for tourist with the experimental results obtained thus far and the applicability of our results in other contexts. Then, we mention about the limitations of this research.

*6.1. Discussion*

The duration of stay ratio and the check-in ratio by information demand level are shown in Section 5.1. We found that setting the rewards according to the information demand level and displaying the rewards with different colors on the map in both missions persuades the tourists to travel to the spots with high information demand level. In other words, it is suggested that the

Variable reward can induce tourist behavior changes. This result suggests that it is possible to not only collect necessary tourism information efficiently, but also to control tourist behavior by gamification. One application example that uses this result is to dynamically change rewards according to the degree of congestion. This will affect the decision of the next destination of the tourist and will be useful for supporting the optimization of the congestion degree of the whole tourist attraction.

Next, the post-survey results on the priority of sightseeing and mission were shown in Section 5.3. These results indicate that tourists are more likely to prioritize missions in the case of the Check-in Mission. A strong tendency to prioritize missions can be interpreted as essentially tourists not enjoying sightseeing. These results suggest that while Check-in Missions are an effective way to gather specific information, the participants tend to focus more on clearing the mission than on enjoying sightseeing. In addition, although there was no significant difference in the effect due to the difference in user types, it was found that the priority differs for each user type. On the other hands, the following comments were obtained as some examples from the free description in post-survey: "It was able to visit places that I don't usually go.", "It could know sightseeing attractions that I didn't know." and "It gives me the motivation to go to places I hadn't planned in advance." That is, although our system was not intended to increase tourism satisfaction, the use of our system resulted in increased tourism enjoyment as a side effect.

Third, we checked the number of photographs and comments received through Free posting and Check-in Missions. We found that approximately 70% of the photos and comments were posted through free posting alone without the application requesting the photographs and comments to be posted as a requirement of the Check-in Mission. Therefore, we believe that an Area Mission and Free posting should generally be adopted for the gamified participatory sensing in tourism, considering the degree of tourist satisfaction. Moreover we suggest that if specific information is urgently needed, a Check-in Mission should be used.

Fourth, we discuss the applicability of these results in other contexts. This approach is basically designed for urban tourism where there are several sightseeing attractions within easy reach by foot. Therefore, we believe that this approach is applicable to all kinds of urban tourism, regardless of the genre of tourist attractions, such as historical sites in Japan or in other countries. On the other hand, travel with high financial and time costs by a car or a train occurs in the case of sightseeing in a natural area or a circular tour. Since it cannot be said that this approach has the influence to promote behavioral changes that exceed the cost, it is considered to be out of adoption. In addition, since this experiment is based on the assumption of a single person's tourism, so we believe that this approach is applicable for single tourists. In the case of a large group of tourists, it is difficult to apply this approach directly, because of other burdens in the decision-making process, such as consensus for next destination determination.

Finally, we discuss the possibility of introducing other gamification elements and point to notice. In this study, we adopted a simple point-reward mission design because there has not been much research on efficient data collection in participatory sensing for tourists. Then, we explore the possibility of introducing other gamification elements based on the obtained results. The first one is *Ranking* or *Leaderboard*. It is generally used as a competitive element in participatory sensing in other domains and was also incorporated in this experiment [54]. However, we received positive feedback that "I was able to compete with other users and enjoyed it", but we also received negative opinion that "I felt rushed due to the rank." Therefore, it is necessary to pay attention to the introduction of ranking in tourism situations. Next is *Meaningful stories*. It is mainly used for the purpose of acquiring knowledge in tourist spots with a historical background such as heritage [53]. Considering the aspect of tourism information collection, one example would be to show participants why the collection is necessary and how useful the collected data is to other users. These may encourage some participants to change their behaviors and may motivate them to collect data. Last is *Full-fldged games (Augmented Reality)*. It has been attracting attention as one of the new tourist contents and encourageing the enjoyment of

tourism [57]. However, even simple elements such as check-in mission have lowered the priority of tourism, so it is necessary to pay attention to the content design.

*6.2. Limitations*

The first limitation of this study is the lack of a control group in our experiment. We wanted to focus on evaluating the impact of different types of gamification on tourist behavior rather than on evaluating the impact of the presence or absence of gamification. The effectiveness of gamification has been sufficiently shown in prior research, as such we assumed gamification would have a positive impact without the need to compare it to a control group.

The second limitation is the amount and homogeneity of the participants. The participants of this experiment were mostly Japanese university students of approximately 20 years in age, although the participants included four non-Japanese and two participants of more than 30 years in age. Therefore, this experimental results cannot be generalized immediately, so we should carry out the experiment in a wild environment with participants of various ages and nationalities.

The third limitation is that, we set the change in demand level manually in the case of a Dynamic Variable reward in the experiment. The reward mechanism was introduced in order to see that the participants can adapt to and follow temporal changes. As a result, we could confirm that the participants adapt to and follow the changes and the reward mechanism can induce a behavior change in participants. Creating a Dynamic Variable reward model using, e.g., the number of active users participating, the fundamental information demand level, and the temporal amount of collected information, is considered useful to solve the problem. However, this is beyond the scope of the present study.

## 7. Conclusions

In this paper, we adopt gamification for participatory sensing in order to efficiently collect dynamic tourism information while taking into account the burden on tourists. Moreover, we investigate the effect of gamification on tourist behavior and tourist satisfaction through an experimental case study. We designed two types of missions (i.e., sensing task), Area Mission and Check-in Mission, and three types of reward mechanisms, Fixed reward, Variable reward, and Dynamic Variable reward. In addition, we set up three types of user type, Sightseeing type, Game type and Reward type, based on the motivation to participation through pre-survey. We integrated these elements into our participatory sensing platform application and conducted in a real-world sightseeing experiment, with 33 participants in Kyoto, Japan. Throughout the experiment, tourism information (e.g., photographs and comments on sightseeing attractions), sensor data and user location data, and answers to the post-survey were collected. We evaluated the effect of gamification on tourist behavior (i.e., data collection efficiency) and satisfaction quantitatively and qualitatively using these collected data. The following are our primary findings. First, we found that Variable reward and Dynamic Variable reward can induce a change in tourist behavior for efficient data collection in both missions. Second, the participants tended to prioritize sightseeing over the Check-in Mission, which can induce a behavior change, but might impact sightseeing enjoyment. Summarizing to answer our research question, we believe that an Area Mission with a Variable reward and Free posting should be adopted basically in gamified participatory sensing for tourists, considering tourist satisfaction and a mechanism to switch to a Check-in Mission, which has a strong influence on behavior change, should be introduced when the demand level becomes extremely high.

In this paper, we could not find a significant difference between subgroups of user type (sightseeing, game and reward). In the future, we will expand to general tourists and conduct experiments with participants of various demographics (gender, age and nationality) to confirm differences due to user-type subgroup and demographic results.

**Author Contributions:** Conceptualization, S.K. and Y.M.; methodology, S.K. and Y.M.; software, S.K. and Y.M.; validation, S.K. and Y.M.; formal analysis, S.K. and H.S.; investigation, S.K. and Y.M.; resources, S.K. and Y.M.; data curation, S.K. and H.S.; writing—original draft preparation, S.K. and M.F.; writing—review and editing, Y.M., H.S., Y.A. and K.Y.; visualization, S.K.; supervision, Y.A. and K.Y.; project administration, S.K. and Y.M; funding acquisition, S.K. and K.Y. All authors have read and agreed to the published version of the manuscript.

**Funding:** This work was supported by JSPS KAKENHI Grant Numbers JP16H01721 and JP18J23281.

**Conflicts of Interest:** The authors declare no conflict of interest. The founding sponsors had no role in the design of the study; in the collection, analyses, or interpretation of the data; in the writing of the manuscript; or in the decision to publish the results.

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
