# Peer review of "Gamified Participatory Sensing in Tourism: An Experimental Study of the Effects on Tourist Behavior and Satisfaction"

_smartcities, doi:10.3390/smartcities3030037_

Round 1
Reviewer 1 Report
The authors present their research on the use of gamification in tourism to obtain crowdsourced data. In order to test their methods, they made a case study with 33 participants in Kyoto.
Overall, I think it is a valuable paper. As the authors themselves mention, there is indeed a lot of gamification in tourism nowadays, but this is not a well studied topic in academia. In this sense, it is a very welcome and timely topic.
However, I think the study leaves much to be desired and the paper needs a lot more information than is currently given.
Mainly, the aims of the study are not well defined (we do not know exactly what information the authors are trying to collect or why), and it is therefore difficult to gauge whether the techniques that are used are appropriate (or could be better obtained through other means), whether the tourism context needs different information than other contexts (eg crowds in mobility), or whether the results of the study really are satisfactory.
I must also say that the game aspect here is not very well developed. I am not sure if the participants are getting anything other than money (eg enjoyment or knowledge) here, or at least anything extra compared to a standard sightseeing experience.
Then, the related work is not presented in the context of the current work. Other tourism gamification examples seem to use some of the same techniques used by the authors. This needs further analysis and discussion.
Finally, the two types of missions are not well justified or described in the paper. Why two? What aims do they achieve? What are the pros and cons?
Summarising the main points, I would like to ask the authors to formulate clearly what they're trying to achieve here and why, to justify why gamification is a promising approach for this, and to evaluate their results in terms of their defined goals.
Reviewer 2 Report
Dear Authors I read your paper, and altough id is really far from my field experience I find it interesting. nevertheless I have some concerns that in my opionion require a major revision process.
I do have some doubt about the experimental design of the research: you attest that there are 33 participants to the experiment (divided into 3 groups of 11). Seems to me a very small number for a complex analysis as the one you are describing. I will suggest you must collect more data, but if you can demonstrate (using the literature ) that this is a reasonable number I can accept it.
The other point is the sample: you tested this in Japan, with Japanese people.
Can you add in the discussion/conclusion some sentence that your approach will work 8or not9 in other contexts? I means natural areas, or historical sites out of Japan, or with different tourists?
Reviewer 3 Report
The paper presents an interesting experiment in how the gamification modifies the behavior of the tourists, taking into consideration especially the reward aspect.
Some small changes may improve the paper:
- On page 5 CGM has not been described before
- Is it possible to have the English translation of Japanese writing as well? It can be useful in figure 3.
- Is it possible to have more details on the degree of congestion (line 222 and around)?
- It is not clear the assignments of participants to groups (lines 303 and following)
- The results section could be better clarified
I suggest accepting the paper with minor revision.
Round 2
Reviewer 1 Report
I appreciate the authors' earnest efforts to improve the paper and to respond to my comments. I am very pleased with the all changes that have been done.
To be honest, I do have one remaining concern. I think that it's somewhat difficult to argue that this is a gamified application when the users do not get a clear benefit from it. In games, it is usually entertainment/enjoyment, so I would hope that a larger focus can be put to make the application pleasant in a future release.